# Hypergraph clustering using Ricci curvature: an edge transport perspective

## Abstract

In this paper, we introduce a novel method for extending Ricci flow to hypergraphs by defining probability measures on the edges and transporting them on the line expansion. This approach yields a new weighting on the edges, which proves particularly effective for community detection. We extensively compare this method with a similar notion of Ricci flow defined on the clique expansion, demonstrating its enhanced sensitivity to the hypergraph structure, especially in the presence of large hyperedges. The two methods are complementary and together form a powerful and highly interpretable framework for community detection in hypergraphs.

## 1 Introduction

Hypergraphs are generalizations of graphs extending the concept of edges to encompass relationships involving any number of vertices rather than being restricted to pairs. Hypergraphs provide a more faithful data representation than graphs as many real-life interactions occur at a group level rather than at a binary one. Examples include social interactions, simultaneous participation at an event, co-authorship, mutual evolutionary traits, spatial proximity or molecular interactions. These interactions are better encoded by hypergraphs instead of traditional graphs, see Davis (1941); Dotson et al. (2022); Konstantinova and Skorobogatov (2001); Torres et al. (2021); Klamt et al. (2009). We refer to Berge (1984) and Bretto (2013) for a comprehensive overview of hypergraph theory.

Recently, machine learning tasks on hypergraphs have encountered a growing success, in particular for classification (Feng et al., 2019), regression (Liu et al., 2021), anomaly detection (Lee et al., 2022) or embedding (Zhang et al., 2019; Antelmi et al., 2023). The present article focuses on the clustering problem or community detection, i.e. recovering labels on the nodes in a fully unsupervised setting from a single hypergraph instance. Several approaches have been taken to tackle this problem such as embeddings using neural networks (Zhang et al., 2021; Lee and Shin, 2023), random walks-based embeddings (Hayashi et al., 2020; Huang et al., 2019) followed by a clustering in the Euclidean space. Some methods developed in Kumar et al. (2020); Kamiński et al. (2024) aim at maximizing hypergraph-modularity functions that measure the strength of a clustering. Finally, let us mention the work of Eyubov et al. (2023) that finds a partition in linear time by streaming all the nodes one by one.

The present work investigates generalizations of Ricci flow based clustering on hypergraphs. Ollivier-Ricci curvature, introduced by Ollivier (2007) for metric spaces and later adapted to graphs (Lin et al., 2011), defines a distance on graph edges that quantifies local curvature using optimal transport theory. Edges within strongly connected communities have a positive curvature while edges bridging two communities are negatively curved. Ricci curvature can be turned into a flow dynamic in order to further stress out this community structure and take some more global graph properties into account. Ricci curvature on graphs has been used to derive theoretical bounds on the spectrum of the graph Laplacian by Lin and Yau (2010) and Bauer et al. (2011). For applied purposes, it has been used for data exploration (Ni et al., 2015), representation learning (Zhang et al., 2023), topological data analysis (Carriere and Blumberg, 2020; Hacquard and Lebovici, 2024), and in particular clustering (Ni et al., 2019), relying on the property of the Ricci flow to place emphasis on the community structure.

There has been a few attempts to generalize Ollivier-Ricci curvature to hypergraphs, in particular by Coupette et al. (2022). The authors introduce a notion of curvature using random walks on the nodes. As pointed out by Chitra and Raphael (2019), in most practical cases, considering a random walk on the hypergraph is equivalent to a random walk on its *clique expansion* (a graph representation where each hyperedge is replaced by a weighted clique). Doing so is a common way to circumvent the complex hypergraph structure by replacing it with a graph with a similar structure. However, as noted in Chitra and Raphael (2019), this reduction can result in significant information loss, as distinct hypergraphs may share identical clique graphs (Hein et al., 2013). In addition, replacing a single hyperedge between $k$ nodes implies adding up to $\binom{k}{2}$ connections between nodes which can be computationally prohibitive in a network with mutual interactions between many agents. Extensions of Ricci curvature on directed hypergraphs have also been proposed by Eidi and Jost (2020).

The main contribution of this work aims at defining a notion of Ricci curvature on hypergraphs where we consider probability distributions on the edges instead than on the nodes. This approach leverages the *line expansion* of the hypergraph, which is a graph representation where nodes correspond to hyperedges, and edges reflect hyperedges intersections. The line expansion is a common way to represent a hypergraph for learning purposes, see Bandyopadhyay et al. (2020); Yang et al. (2022). In addition, it has been demonstrated in Kirkland (2018) that almost all the hypergraph information is retained by the joint knowledge of both its clique and line expansions. The current work puts a particular focus on clustering. We perform a thorough experimental study on synthetic and real data comparing the approach where we transport measures on the edges to the more standard approach where we transport measures on the nodes. More precisely, this notion of edge transport should be favored when we have a large number of small communities, when the edges between communities have a larger cardinality than the ones within communities, and more generally for hypergraphs with very large hyperedges as this approach is much more efficient computationally.

The work is organized as follows: Section 2 introduces the foundational concepts of hypergraph theory and Ollivier-Ricci curvature on graphs. Section 3 presents two different possible expansions of Ricci curvature for hypergraphs: the standard one where we transport measures on the nodes, and the main contribution of this paper where we transport measures on the edges. We also compare the two approaches theoretically on a synthetic example. Section 4 provides a detailed baseline of experiments on synthetic data and on real data where we compare both methods to state-of-the-art clustering algorithms on hypergraphs, along with a computational analysis of both methods.

## 2 Model

This section presents the main concepts related to hypergraph analysis and the hypergraph partitioning problem. We also investigate standard methods to replace hypergraphs by traditional graphs and the loss of information it implies.

### 2.1 Hypergraphs and associated graphs

**Hypergraphs, definitions** *Undirected hypergraphs*, sometimes referred to as *hypernetworks* or *2-modes networks*, consist of a set of nodes $V$ and a set of *hyperedges* $E$. Extending the concept of graphs where edges link two distinct nodes, hyperedges are defined as non-empty sets of nodes of arbitrary size. For a hyperedge $e = (u_1, \ldots, u_k)$, its cardinality $k$ is referred to as its *size*. A hypergraph in which all hyperedges have the same size $k$ is called a $k-$*uniform hypergraph*. Notably, graphs are $2-$uniform hypergraphs. We further define the star of a vertex $v \in V$ as the set of hyperedges that include $v$: $St(v) = \{e \in E | v \in e\}$.

**Clique expansion** Given a hypergraph $H$ with node set $V = (v_1, \ldots, v_n)$ and edge set $E = (e_1, \ldots, e_m)$, its *incidence matrix* $\mathbf{I} \in \{0,1\}^{n \times m}$ is defined as $\mathbf{I}_{i,j} = 1$ if $v_i \in e_j$ and 0 otherwise. This representation as a rectangular matrix makes further analysis of hypergraphs much more complicated than that of graphs, which can be described by their adjacency matrix. A commonly used simplification of hypergraphs is the *clique expansion*, where each edge is replaced by a clique. More precisely, the clique expansion $\mathbf{C}(H)$ of the hypergraph $H$ is the graph with node set $V$ and with an edge $(x, y)$ if there exists a hyperedge $e \in E$ such that $x, y \in e$. The clique expansion provides a convenient simplification of hypergraphs by grouping

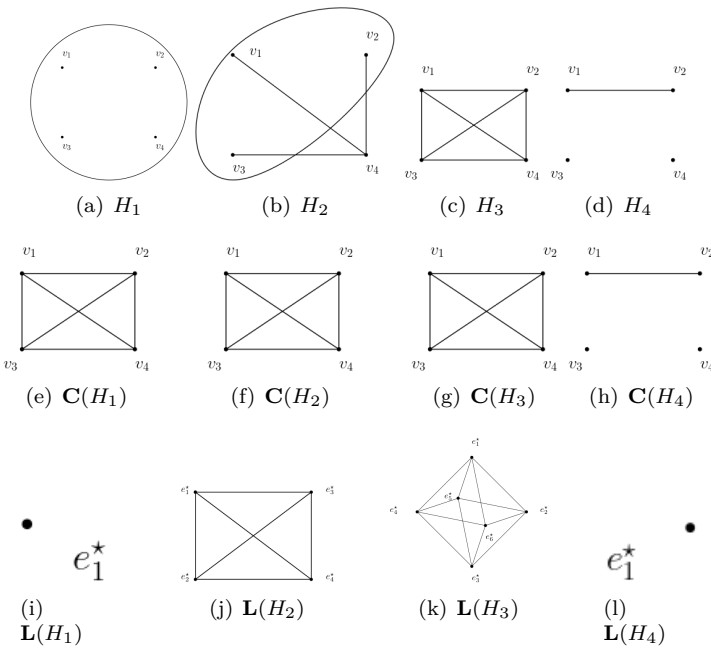

Figure 1: Examples of hypergraphs $H_i$ with their clique $\mathbf{C}(H_i)$ and line $\mathbf{L}(H_i)$ expansions. $H_1$, $H_2$ and $H_3$ share the same clique but have different line expansions.

nodes belonging to the same edge. However, there is a certain loss of information since multiple hypergraphs share the same clique expansion, see Figure 1. We refer to Hein et al. (2013) for a more thorough analysis of differences between hypergraphs and their clique expansion. Notably, according to Chitra and Raphael (2019) and Hayashi et al. (2020), unless the nodes are assined an *edge-dependent weighting*, standard notions of random walks and Laplacians on the hypergraphs can be expressed on a reweighted clique expansion. This implies that directly applying Laplacian-based methods, such as clustering, on hypergraphs often incurs the information loss inherent in their clique expansions. We note that the adjacency matrix $A_C$ of the clique graph can be derived from the incidence matrix via $A_C = \mathbf{II}^T - D_V$ where $D_v$ is the diagonal node-degree matrix, with entries $d_i = \sum_{j=1}^{n} \mathbf{I}_{i,j}$.

**Line expansion**    Another graph associated with $H$ is the *line graph* $\mathbf{L}(H)$. Each edge in $H$ now corresponds to a node in $\mathbf{L}(H)$. Thus, $\mathbf{L}(H)$ has node set $(e_1^\star, \ldots, e_m^\star)$ and we have an edge $(e_i^\star, e_j^\star)$ in $\mathbf{L}(H)$ if the corresponding hyperedges intersect in $H$, i.e. $e_i \cap e_j \neq \emptyset$. Similarly to the clique expansion, many hypergraphs share the same line expansion, see Figure 1. However, joint knowledge of both the line and clique expansions seems to capture most of the structural information of the hypergraph. Indeed in Figure 1, hypergraphs $H_1$, $H_2$ and $H_3$ all share the same clique expansion while having different line expansions (they even have different node sets). Similarly to the clique graph, the adjacency matrix $A_L$ of the line graph can be derived from the incidence matrix via $A_L = \mathbf{I}^T \mathbf{I} - D_e$ where $D_e$ is the diagonal matrix of edge-lengths, with entries $\delta_i = |e_i|$. Let us define the dual hypergraph $H^\star = (E^\star, V^\star)$ as the hypergraph obtained by swapping the edge and node sets, i.e. $E^\star = (e_1^\star, \ldots, e_m^\star)$ and $V^\star = (v_1^\star, \ldots v_n^\star)$ where $v_i^\star = \{e_j^\star | v_i \in e_j\}$. We can easily observe, see Zhou et al. (2022), that the clique expansion is the line graph of the dual hypergraph: $\mathbf{C}(H) = \mathbf{L}(H^\star)$.

**Gram mates**    We can wonder if the joint knowledge of both the clique graph and the line graph allows to reconstruct the hypergraph. The answer is negative, as there exists pairs of distinct 0-1 matrices $(A, B)$ such that $AA^T = BB^T$ and $A^T A = B^T B$. Such matrices are called *Gram mates*, and we refer to Kim and Kirkland (2022) for a proof of this result and a thorough analysis of families of matrices that generate Gram mates. However, it has been demonstrated in Corollary 1.1.1 of Kirkland (2018) that if we uniformly pick two $m \times n$ 0-1 matrices, the probability that they are Gram mates decays exponentially as $(m, n) \to \infty$. Thus, for practical applications, the information loss from replacing a hypergraph with its clique and line expansions together is negligible. This representation also simplifies hypergraph analysis significantly.

**Clustering** We consider hypergraphs where nodes are associated with discrete labels $\{1, \ldots, K\}$, representing communities or clusters. Intuitively, such hypergraphs exhibit a community structure, with denser connections between nodes sharing the same label compared to those with different labels. We assume a fully unsupervised setting, where we try to infer the labels (up to permutation) simply via the hypergraph structure. This generalization of the graph partitioning problem has been studied extensively, and we refer to Çatalyürek et al. (2023) for a survey of different approaches to tackle this problem. A popular method to determine the quality of a partitioning is given by *modularity* maximization, initially defined by Newman (2006). We recall its definition in the graph case. For a graph $G = (V, E)$ and a partitioning $\mathbf{C} = (C_1, \ldots, C_K)$ of $V$ into $K$ subsets, we define the *modularity function $Q$* as:

$$Q(\mathbf{C}) = \sum_{i=1}^{K} \frac{e_G(C_i)}{|E|} - \sum_{i=1}^{K} \left( \frac{vol(C_i)}{vol(V)} \right)^2,$$

where $e_G(C_i)$ is the number of edges in the subgraph of $G$ induced by a node set $C_i$, and $vol(A) = \sum_{v \in A} deg(v)$ is the *volume* of a subset $A$ of nodes. Up to renormalization, the modularity function computes the difference between the number of edges uncut by the partitioning and the expected number of edges the same partitioning would yield in the Chung-Lu random graph model introduced in Chung and Lu (2006). Maximizing $Q$ over every partitioning yields partitions that minimize cuts within communities, aligning with the underlying structure. There have been several works generalizing modularity for hypergraphs, notably Kumar et al. (2020) and Kamiński et al. (2024), by generalizing the Chung-Lu random model to hypergraphs, and allowing different weights to hyperedges of different sizes. Modularity also serves as a ground truth-free measure of the quality of a given clustering.

## 2.2 Ricci curvature and flows on graphs

This paper addresses the clustering problem using an extension of Ollivier-Ricci curvature to hypergraphs. This method extends prior works on Ollivier-Ricci curvature, initially defined in Ollivier (2007) on metric spaces, and extended to the case of graphs in Lin et al. (2011). In Ni et al. (2019), the Ricci curvature is turned into a discrete flow such that edges between communities are heavily curved. They directly derive a clustering algorithm. We start by recalling the key concepts of their method in the case of graphs.

**Discrete Ricci curvature** Let $G = (V, E)$ be a weighted graph with a weight function $w : E \to \mathbb{R}_{\geq 0}$. In what follows, the function $w$ is set to be a *dissimilarity measure* between nodes. Let $d$ denote the shortest path distance between nodes induced by weights $w$. For each node $x$, we define a probability measure $\nu_x$ over $x$ and its neighbors given by:

$$\nu_x(y) = \begin{cases} \alpha & \text{if } y = x, \\ \frac{1-\alpha}{|\mathcal{N}(x)|} & \text{if } y \in \mathcal{N}(x), \\ 0 & \text{otherwise,} \end{cases} \tag{1}$$

where $\mathcal{N}(x)$ denotes the set of neighbors of $x$ and $\alpha \in [0, 1]$ is a real parameter controlling the amount of mass retained at $x$. More general measures on the nodes are considered in Ni et al. (2019).

The *discrete Ricci curvature $\kappa$* is a function defined on the edges of the graph such that for an edge $e = (x, y)$:

$$\kappa(x, y) = 1 - \frac{W(\nu_x, \nu_y)}{d(x, y)},$$

where $W$ is the Wasserstein distance between probability measures for the cost function $d$, see Santambrogio (2015). More precisely, a discrete transport plan $T$ between the measures $\nu_x$ and $\nu_y$ is an application $T : V \times V \to [0, 1]$ such that $\sum_{v' \in V} A(u, v') = \nu_x(u)$ and $\sum_{u' \in V} A(u', v) = \nu_y(v)$. The Wasserstein distance corresponds to the total cost of moving $\nu_x$ to $\nu_y$ with the optimal transport plan for a cost function $d$, i.e.

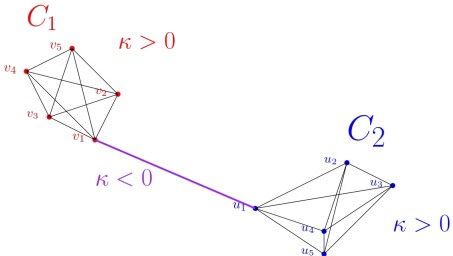

Figure 2: Ollivier-Ricci curvature on a graph with a clear two-communities structure.

$$W(\nu_x, \nu_y) = \inf \left\{ \sum_{u,v \in V} T(u,v)d(u,v) | T \text{ transport plan between } \nu_x \text{ and } \nu_y \right\}.$$

The sign of $\kappa$ provides information on the density of connections. Indeed, two nodes $x$ and $y$ belonging to the same community will typically share many neighbors, hence transporting $\nu_x$ to $\nu_y$ has a smaller cost than moving $x$ to $y$ and therefore induces a positive curvature. Conversely, nodes belonging to different communities do not share many neighbors, such that moving $\nu_x$ to $\nu_y$ requires traveling through the edge $(x,y)$ bridging the two communities. This in turn corresponds to a negative curvature. We refer to Figure 2 for a concrete example: when transporting $\nu_{v_i}$ to $\nu_{v_j}$ and $\nu_{u_i}$ to $\nu_{u_j}$, most of the mass remains on the nodes such that measures are transported at a low cost. In the extreme case where $\alpha = 1/5$, all the mass remains and $\kappa(v_i, v_j) = \kappa(u_i, u_j) = 1$. When transporting $\nu_{v_1}$ to $\nu_{u_1}$, all the $v_i$'s must be transported to $u_i$'s through the edge $(u_1, v_1)$ such that the total transport cost is larger than $d(u_1, v_1)$ and therefore $\kappa(u_1, v_1) < 0$.

It instantly appears that trimming edges of low curvature naturally partitions the graph where each connected component corresponds to a different community. We note that Ricci curvature can theoretically be computed for any pair of nodes in the graph. However, for our purposes, we only need to compute it for adjacent nodes.

**Discrete Ricci flow and community detection** To amplify the impact of heavily negatively curved edges and incorporate more global graph structure, we turn Ricci curvature into a discrete flow that modifies the graph weights. We recall that the edge weights are dissimilarities, and a high weight corresponds to a high distance between nodes. The initial weight between $x$ and $y$ is $w^{(0)}(x,y) = w(x,y)$. We iteratively update the weights with:

$$w^{(l+1)}(x,y) = \left(1 - \kappa^{(l)}(x,y)\right) w^{(l)}(x,y), \tag{2}$$

where $\kappa^{(l)}$ is the Ricci curvature computed with the distance induced by the weights $w^{(l)}$. The updated weights after the $l$-th iteration are referred to as the *Ricci flow*. This flow dynamic stretches edges of low curvature, making it more and more costly to transport measures between communities. The complete clustering algorithm is described by Algorithm 1, and simply trims edges with a high Ricci flow. The choice of the number of flow iterations $N$ is mostly heuristic, we refer to (Ni et al., 2019) and the corresponding `python` library `GraphRicciCurvature` for some guidelines. The threshold $\tau$ at which to trim the weights is a crucial parameter and can also be taken as the one that maximizes the *modularity function* defined in Section 2.1, which evaluates clustering quality without ground truth labels.

This approach ensures that inter-community connections are down-weighted while preserving intra-community coherence, enabling a robust and interpretable clustering.

## 3 Two complementary notions of Ricci curvature on hypergraphs

Extending Ricci curvature to hypergraphs presents two main challenges:

- Ricci curvature is traditionally defined for pairs of nodes and needs to be extended to hyperedges.

**Input:** Graph $G = (V, E)$, Number of iterations $N$, Threshold $\tau$
**Output:** Clustering labels $Y$
**for** $i \leftarrow 1$ **to** $N$ **do**
   |   Update the weights using Equation 2;
**end**
Trim all edges with a weight larger than $\tau$ to create a new graph $\tilde{G}$ ;
Compute connected components $C_1 \coprod \ldots \coprod C_k$ of $\tilde{G}$ ;
**for** $i \leftarrow 1$ **to** $|V|$ **do**
   |   Set $Y_i = j$ if $v_i \in C_j$;
**end**

**Algorithm 1:** Ricci flow algorithm.

- Computing transport distances between measures defined on the nodes of a hypergraph typically relies on the clique expansion (see Section 2.1 and Hayashi et al. (2020)). As a consequence, this approach introduces a loss of structural information, as clique expansions fail to capture certain higher-order interactions.

The first challenge is straightforward to treat, for instance by taking the average or the maximum of all pairwise curvatures of nodes in a hyperedge (as in Coupette et al. (2022)). We will discuss different possibilities in Section 3.1. The second challenge constitutes the main contribution of this paper and will further be developed in Section 3.2 where we explore a method to transport hyperedges.

Note that we restrict our analysis to the Ollivier-Ricci curvature. Another notion, the *Forman-Ricci curvature* (Forman, 2003), has been extended to the case of hypergraphs in Leal et al. (2021). We refer to Samal et al. (2018) for a detailed comparison of these two curvature types in the graph case.

### 3.1 Nodes transport on the clique graph

**Clique graph transport** Let $H$ be a hypergraph with nodes $V = (v_1, \ldots, v_n)$ and edges $E = (e_1, \ldots, e_m)$. To compute the Ricci curvature between two nodes $x$ and $y$, we can still define measures $\nu_x$ and $\nu_y$ on $x$ and $y$ and their neighbors following Equation 1, as the definition of a neighbor is unchanged. The first solution is simply to replace the hypergraph with its clique graph. Two possibilities are considered to obtain a clique graph from a hypergraph. For the sake of clarity, we assume the hypergraph to be unweighted. Let $A_C = (a_{ij})_{1 \leq i,j \leq n}$ be the adjacency matrix of the clique expansion of $H$.

(i) The first option is to consider a simple unweighted clique graph expansion where

$$a_{ij} = \begin{cases} 1 & \text{if there exists } e \in H \text{ such that } (v_i, v_j) \subset e, \\ 0 & \text{otherwise.} \end{cases}$$

(ii) The second option is proposed by Zhou et al. (2022) and relies on a Jaccard index weighting. Recall that for two finite sets $U$ and $V$, the Jaccard index $J(U, V) = \frac{|U \cap V|}{|U \cup V|}$ is a measure of similarity between sets $U$ and $V$. Let $H^\star = (V^\star, E^\star)$ be the dual hypergraph as defined in 2.1. We remark that for two nodes $v_i$ and $v_j$, $a_{ij} = 0$ if and only if $|v_i^\star \cap v_j^\star| = 0$. The weight of two adjacent nodes in the clique graph is given by $1/J(v_i^\star, v_j^\star)$. The J<accard index is inverted to provide a dissimilarity weighting.

**Aggregating pairwise curvatures** After computing the Ricci flow $w_{\mathcal{N}}^{(l)}(e)$ for every pair of adjacent nodes in the clique graph of $H$, the Ricci weight of a given hyperedge $e \in E$ is obtained by aggregating all pairwise flows between nodes of $e$:

$$w_{\mathcal{N}}^{(l)}(e) = \underset{v_i, v_j \in e}{Agg} \left( w^{(l)}(v_i, v_j) \right),$$

where $Agg$ is an aggregation function. Experimentally, we found that aggregating using the *maximum* of all pairwise curvatures systematically yielded the best clustering performance.

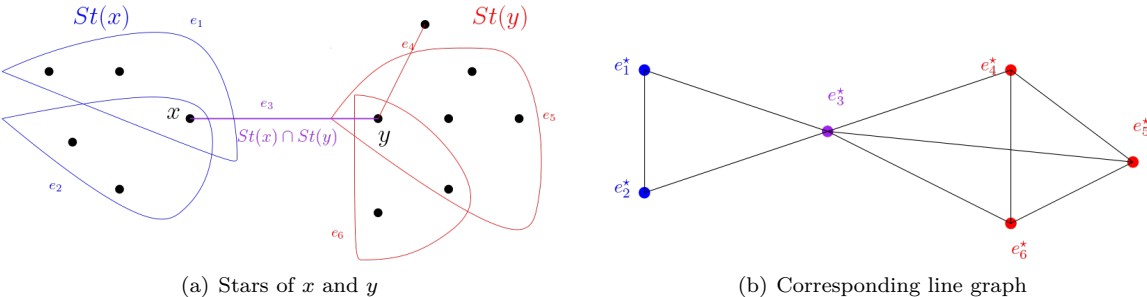

(a) Stars of $x$ and $y$         (b) Corresponding line graph

Figure 3: Edge-Ricci transport between nodes $x$ and $y$. A measure on $St(x)$ is transported onto a measure on $St(y)$ via the line graph.

The clustering pipeline given by Algorithm 1 then operates similarly by trimming all hyperedges with a Ricci flow larger than some threshold. The optimal threshold can be found by either maximizing the modularity of the clique graph, or a corresponding notion of *hypergraph modularity*, see Kamiński et al. (2024).

## 3.2 Edges transport on the line graph

As discussed in 2.1, replacing a hypergraph by its clique graph induces a substantial loss of information. To address this, we propose a novel notion of Ricci curvature specific to hypergraphs. Instead of transporting measures defined on the nodes of a hypergraph, we transport measures defined on its edges.

**Edge-Ricci curvature** The proposed method also computes pairwise curvature and then extends it to hyperedges using an aggregation step similar to the one previously defined. We first describe the Ricci curvature computation for a hypergraph $H = (V, E)$ with dissimilarity weights $d : E \to \mathbb{R}_{>0}$. Let $x$ and $y$ be two adjacent nodes in the hypergraph. We define probability measures $\mu_x$ and $\mu_y$ on their stars $St(x)$ and $St(y)$, see Figure 3. The most common choice is to consider $\mu_x$ and $\mu_y$ to be uniform probability measures. Similarly to what is commonly done for standard Ricci curvature on graphs, another option is given a parameter $\alpha \in [0, 1]$, to define the following measure:

$$
\mu_x(e) = \begin{cases} \frac{\alpha d(e)}{C_1} & \text{if } x, y \in e \\ \frac{(1-\alpha)d(e)}{C_2} & \text{if } e \in St(x) \text{ and } e \notin St(y) \\ 0 & \text{otherwise,} \end{cases} \tag{3}
$$

where $C_1 = \sum_{e \in St(x) \cap St(y)} d(e)$ and $C_2 = \sum_{e \in St(x) \setminus St(y)} d(e)$ are normalization constants to ensure $\mu_x$ is a probability measure. The definition of $\mu_y$ is analogous. The distance $d_H$ between two edges $e$ and $f$ is defined using the line graph: $d_H(e, f) = d_{LG}(e^\star, f^\star)$ where $d_{LG}$ is the standard shortest path distance for graphs. Note that similarly to the clique graph, there are several ways the line graph can be weighted (for instance using intersection or Jaccard weights between edges). This in turn ensures that similarly to the graph case, we can define an optimal transport plan and a corresponding Wasserstein distance $W$ between the two measures $\mu_x$ and $\mu_y$. The *edge-Ricci curvature* $\kappa_{\mathcal{E}}$ between two nodes $x$ and $y$ is therefore defined as

$$
\kappa_{\mathcal{E}}(x, y) = 1 - \frac{W(\mu_x, \mu_y)}{\max\limits_{e \in St(x) \cap St(y)} d(e)}.
$$

Pairs of nodes in different communities tend to have very different stars, which implies in turn a low curvature. Conversely, if the two nodes $x$ and $y$ belong to the same community, the measures $\mu_x$ and $\mu_y$ can be transported onto each other with a low cost, resulting in a curvature close to 1. We remark that even in the case of a 2-uniform hypergraph, this definition is not equivalent to the usual curvature on graphs from Section 2.2. In addition, we lose the interpretability observed in Figure 2 concerning the sign of the curvature.

**Edge-Ricci flow** Similarly to the graph case, the curvature is then turned into a discrete flow. We start with an initial edge weighting $w_{\mathcal{E}}^{(0)} = d$. At the $l$-th iteration, the weights are updated according to the following three steps:

(i) Compute the Ricci curvature $\kappa_{\mathcal{E}}^{(l)}$ for every pair of adjacent nodes in the weighted hypergraph.

(ii) For every hyperedge $e$, aggregate all pairwise curvatures with an aggregation function $Agg$ to define the curvature of the hyperedge, similarly to Section 3.1:

$$\kappa_{\mathcal{E}}^{(l)}(e) = \underset{v_i, v_j \in e}{Agg} \left( \kappa_{\mathcal{E}}^{(l)}(v_i, v_j) \right).$$

(iii) For every hyperedge $e$, update the edge weights using the Ricci flow dynamic:

$$w_{\mathcal{E}}^{(l+1)}(e) = \left( 1 - \kappa_{\mathcal{E}}^{(l)}(e) \right) w_{\mathcal{E}}^{(l)}(e).$$

Similarly to the graph case this procedure is iterated $N$ times, and hyperedges with a weight larger than a given threshold are trimmed. The resulting connected components form the assigned node communities. In traditional Ricci flow, negatively curved edges gradually gain weight, increasing the cost of moving probability measures through them. Here, we take a dual perspective: edges with low curvature are assigned increasing mass, making them progressively more expensive to transport. The line graph, however, remains unchanged, such that distances between edges are unaffected by the flow process.

### 3.3 A toy clustering example

We start by comparing the node-Ricci flow from Section 3.1 with the edge-Ricci flow from Section 3.2 in a synthetic example where transport maps and curvatures can be computed explicitly. We adopt a very similar setting to the one of Ni et al. (2019) where we consider a family of hypergraphs $H(a, b)$ where $a, b \geq 3$ are integers. Consider $b$ complete binary graphs $C_1, \ldots, C_b$ on $a$ distinct vertices. For each complete graph $C_i$, take a particular vertex $v_i$ (called gateway vertex) and consider the hyperedge of size $b$, $(v_1, \ldots v_b)$ connecting all gateway vertices across the communities. This hypergraph exhibits a clear community structure where each $C_i$ forms a distinct community, and there is a single hyperedge connecting all the communities together. Figure 4 provides a visual representation. The case of the node-Ricci flow for an unweighted clique expansion has been treated explicitly in Ni et al. (2019), for measures defined by Equation 1 with $\alpha = 0$. They observe that by symmetry, in the clique expansion, the Ricci flow at the $l-$th iteration can only take three possible values: $w_{\mathcal{N},1}^{(l)}$ for edges between two gateway nodes, $w_{\mathcal{N},2}^{(l)}$ for edges between a gateway node and another node from its community, and $w_{\mathcal{N},3}^{(l)}$ for edges between two non-gateway nodes from the same community. They further demonstrate that if $a > b$, $w_{\mathcal{N},3}^{(l)} = \left( \frac{1}{a} \right)^l \underset{l \to \infty}{\to} 0$ and that there exists $\lambda > \frac{1}{a}$ and constants $c_1 > c_2$ such that $w_{\mathcal{N},1}^{(l)} = c_1 \lambda^l + o(\lambda^l)$ and $w_{\mathcal{N},2}^{(l)} = c_2 \lambda^l + o(\lambda^l)$ as $l \to \infty$. This implies that the Ricci flow-based trimming algorithm properly identifies the communities $C_1, \ldots, C_b$. In the case of the edge-Ricci flow, we have the following result:

**Proposition 1** *Keeping the same notation as above and considering the edge-Ricci flow with parameter $\alpha = 0$ and aggregation $Agg$ with the maximum function. At the $l$-th iteration, the Ricci flow can only take three possible values: $w_{\mathcal{N},1}^{(l)}$ for the edge of size $b$ connecting all gateway nodes, $w_{\mathcal{N},2}^{(l)}$ for binary edges between a gateway node and another node from its community, and $w_{\mathcal{N},3}^{(l)}$ for binary edges between two non-gateway nodes from the same community. Under these circumstances, we have that for all $l > 0$:*

$$\begin{cases} w_{\mathcal{N},1}^{(l)} = 2, \\ w_{\mathcal{N},2}^{(l)} = \frac{4 + (a-2) w_{\mathcal{N},2}^{(l-1)}}{2 + (a-2) w_{\mathcal{N},2}^{(l-1)}} \in [1, 2], \\ w_{\mathcal{N},3}^{(l)} = 1. \end{cases}$$

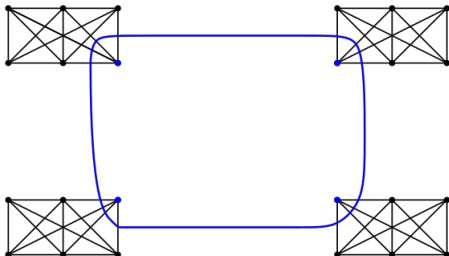

Figure 4: Example of hypergraph $H(a, b)$ with $a = 6$ and $b = 4$. The gateway nodes are represented in blue.

The proof is deferred to Appendix A. Proposition 1 implies that the edge-Ricci flow can also identify the community structure with an edge cut-off parameter between 1 and 2. We notice that in this case, the explicit expression of the Ricci flow, although much simpler than in the node-Ricci case, does not have an exponential decay to 0 for the intra-community edges. This implies that the cut-off parameter might be more delicate to tune in practice. However, note that Proposition 1 holds for any values of $a$ and $b$, contrary to the result of Ni et al. (2019) which only holds for $a > b$. This implies that in practice, edge-Ricci allows for better detection of many small communities.

## 4 Experiments

To evaluate the practical effectiveness of Ricci flow for hypergraph clustering, we conducted a range of experiments on synthetic and real data. We remarked that the choice of the weights in the clique graph and the line graph had little impact on the final accuracy, and opted for a Jaccard type weighting. Curvatures of hyperedges are computed by aggregating curvatures of pairs of nodes using the maximum function. For node-Ricci flow, we have used the python library `GraphRicciCurvature` with default parameters for the measures $\nu$ defined on the nodes. We did not observe a great sensibility to the $\alpha$ parameter for the measures $\mu$ defined on the edges. The clustering accuracy is measured using the *Normalized Mutual Information score* (NMI). The implementations of edge-Ricci and node-Ricci flows for hypergraphs are available at `https://github.com/AnonRicciHG/Ricci_curv_HG` and are directly adapted from the `GraphRicciCurvature` library.

### 4.1 Hypergraphs stochastic block models

We first evaluate our method on a synthetic set-up to highlight differences between edge-Ricci and node-Ricci flows. We consider a hypergraph stochastic block model inspired by Kim et al. (2018). We consider a hypergraph with $n$ nodes and $k$ communities. We give ourselves two size parameters $s_{in}$ and $s_{out}$ and generate $N_{in}$ hyperedges of size $s_{in}$ between nodes of the same community for each community and $N_{out}$ hyperedges of size $s_{out}$ containing at least one node of each community. In the graph case where $s_{in} = s_{out} = 2$, this is a form of stochastic block model, see Abbe (2018). This is a simple model where intra-community edges can be interpreted as signal and inter-community edges as noise. In the hypergraph case, as soon as $s_{in} \geq 3$, inter-community edges are also regrouping nodes of the same community together and cannot simply be interpreted as pure noise.

We try to reconstruct the communities using edge-Ricci and node-Ricci flows clustering algorithms. We consider randomly generated hypergraphs of $n = 100$ nodes, $k = 2$ equal-sized communities and compare the performances of the two methods for various values of $s_{in}$ and $s_{out}$. We fix the value of $N_{in}$ and measure the evolution of the clustering accuracy for a growing number $N_{out}$ of "noisy" hyperedges. We compute $N = 20$ flow iterations for each method and take the thresholding parameter $\tau$ that maximizes the NMI. We average over 10 random hypergraph generations and report the corresponding NMI in Figure 5. We remark that nodes-flow seems to perform comparatively better than edges-flow when the size of intra-community edges $s_{in}$ is large and the size of inter-community edges $s_{out}$ is small. More precisely, based on the numerical results of Figure 5, we can make the heuristic observation that edge-Ricci yields a better clustering performance than node-Ricci whenever $s_{in} < s_{out} - 1$ and that node-Ricci performs better whenever $s_{in} \geq s_{out}$. When $s_{out} = s_{in} + 1$, the two methods have a comparative performance, depending on the level of noise. These differences can be simply understood heuristically: when contaminating a hypergraph by adding a hyperedge

of size $s$, the clique graph will have up to $s(s-1)/2$ new edges while the line graph will only have a single new additional node. This accounts for an improved robustness of edges-flow to large hyperedges contamination, as it mostly relies on shortest-path computations on the line-graph.

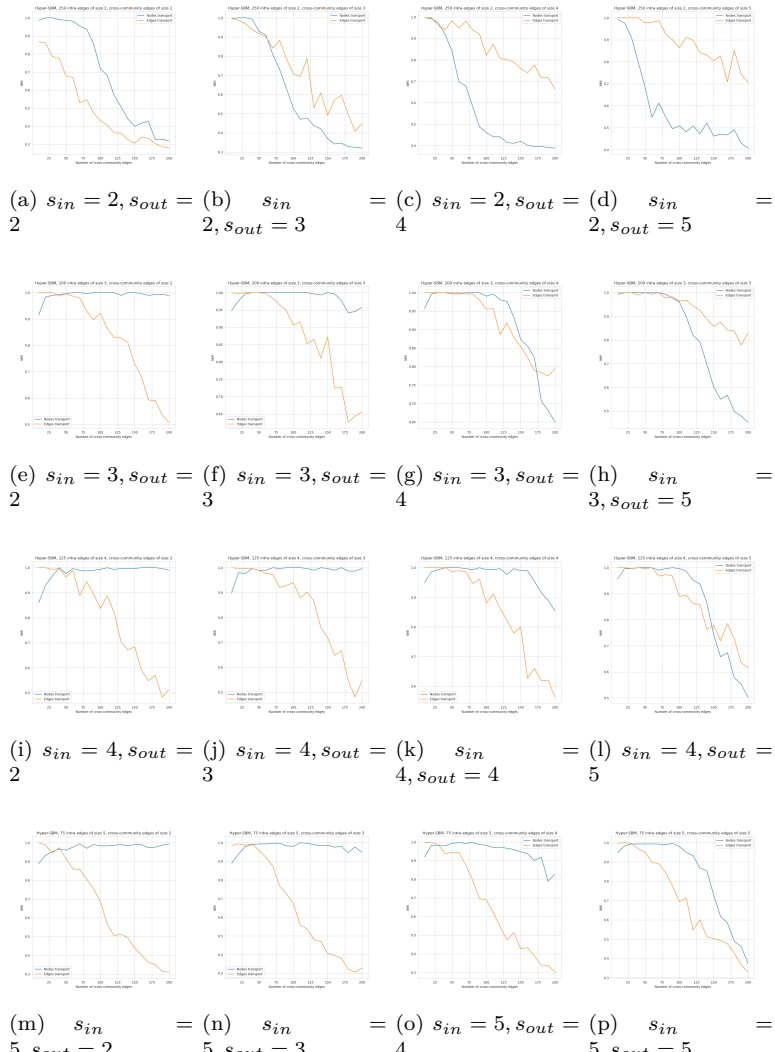

(a) $s_{in} = 2, s_{out} = 2$ (b) $s_{in} = 2, s_{out} = 3$ (c) $s_{in} = 2, s_{out} = 4$ (d) $s_{in} = 2, s_{out} = 5$

(e) $s_{in} = 3, s_{out} = 2$ (f) $s_{in} = 3, s_{out} = 3$ (g) $s_{in} = 3, s_{out} = 4$ (h) $s_{in} = 3, s_{out} = 5$

(i) $s_{in} = 4, s_{out} = 2$ (j) $s_{in} = 4, s_{out} = 3$ (k) $s_{in} = 4, s_{out} = 4$ (l) $s_{in} = 4, s_{out} = 5$

(m) $s_{in} = 5, s_{out} = 2$ (n) $s_{in} = 5, s_{out} = 3$ (o) $s_{in} = 5, s_{out} = 4$ (p) $s_{in} = 5, s_{out} = 5$

Figure 5: Hypergraph stochastic block model reconstruction using hypergraph notions of Ricci flow.

## 4.2 Real data

We further compare the clustering performance of nodes and edges-flow with state-of-the-art graph and hypergraph clustering methods.

**Methodology and data** We evaluate our method on the datasets presented in Table 1. Data and code to reproduce the experiments are available at `https://github.com/AnonRicciHG/Ricci_curv_HG`. We adopt the same benchmark study as in Lee and Shin (2023). For each dataset, we iterate Ricci flows $N = 20$ times. The edge-cutting threshold $\tau$ is taken as the one that maximizes the hypergraph modularity, computed with the `hypernetx` python package. We did not observe any significant change of accuracy regarding the weights of the clique and line expansions, as well as in the value of $\alpha$ in the measures defined in Equations 1 and 3. Pairwise curvatures are aggregated at the hyperedge level using the *maximum* function.

| Dataset | Cora-C | Cora-A | Citeseer | Pubmed | Zoo | Mushroom | NTU2012 |
|---|---|---|---|---|---|---|---|
| ♯ Nodes | 1434 | 2388 | 1458 | 3840 | 101 | 8124 | 2012 |
| ♯ Hyperedges | 1579 | 1072 | 1079 | 7963 | 43 | 298 | 2012 |
| Avg. hyperedge size | 3.0 | 4.3 | 3.2 | 4.4 | 39.9 | 136.3 | 5.0 |
| Avg. node degree | 3.3 | 1.9 | 2.4 | 9.0 | 17.0 | 5.0 | 5.0 |
| ♯ Communities | 7 | 7 | 6 | 3 | 7 | 2 | 67 |

Table 1: Basic statistics of hypergraph data used in the experiments.

| Dataset | Cora-C | Cora-A | Citeseer | Pubmed | Zoo | Mushroom | NTU2012 |
|---|---|---|---|---|---|---|---|
| Node2Vec | 39.1 | 16.0 | 24.5 | 23.1 | 11.5 | 1.6 | 78.3 |
| DGI | **54.8** | 45.2 | 40.1 | 30.4 | 13.0 | OOM | 79.6 |
| GRACE | 44.4 | 37.9 | 33.3 | 16.7 | 7.3 | OOM | 74.6 |
| $S^2$-HHGR | 51.0 | 45.4 | 41.1 | 27.7 | 90.9 | 18.6 | 82.7 |
| TriCL | 54.5 | **49.8** | **44.1** | **30.0** | 91.2 | 3.8 | **83.2** |
| Modularity | 45.0 | 33.4 | 33.8 | 25.0 | 77.7 | **43.4** | 74.5 |
| N-Ricci | 45.8 | 39.4 | 38.8 | 27.8 | 96.2 | OOT | 82.3 |
| E-Ricci | 43.3 | 39.1 | 38.3 | 22.7 | **100.0** | 42.3 | 80.9 |

Table 2: NMI clustering accuracy on real datasets.

**Quantitative results** We report the NMI of our method in Table 2, where we compare ourselves with graph and hypergraph partitioning state-of-the-art methods. Node2vec (Grover and Leskovec, 2016), DGI (Veličković et al., 2018) and GRACE (Zhu et al., 2020) embed the hypergraph's clique expansion in a Euclidean space using various neural networks architectures. $S^2$-HHGR from Zhang et al. (2021) and TriCL from Lee and Shin (2023) directly embed the hypergraphs using a neural network architecture. Communities are then detected using a $k$-means algorithm. Note that for TriCL and $S^2$-HHGR, additional nodes-feature information is used to improve the prediction, while Ricci flow methods relies only on the hypergraph structural information. All scores for these methods are reported from Lee and Shin (2023). We also report the score obtained by the modularity maximization algorithm from Kamiński et al. (2021). OOM indicates an out-of-memory error on a 24GB GPU (reported from Lee and Shin (2023)). OOT indicates that the results could not be obtained within 72 hours on a single 8-core laptop with a Intel(R) Core(TM) i5-8300H CPU @ 2.30GHz processor unit.

Ricci-based clustering methods provide an overall competitive accuracy, with a better performance than simple graph vectorization methods, while managing to be as competitive as state-of-the-art neural network-based hypergraph embeddings on a few datasets. In particular, we achieve a perfect clustering on the `Zoo` dataset using edge-Ricci flow. This dataset stands out for having very few nodes and edges and demonstrates the ability of Ricci-based methods to capture fine structural information on small graphs. We claim that one of the strengths of our method lies in its high level of explainability as opposed to other methods based on neural network embeddings.

### 4.3 Computational comparison

We observe in Table 2 that nodes-Ricci flow failed to provide results for the `Mushroom` dataset in a reasonable time. Even after 72 hours, not a single Ricci-flow iteration was completed, despite this dataset containing a relatively small number of hyperedges (see Table 1). To better understand this limitation, we further investigate the computational complexity of both methods.

Computing the Ricci curvature of all edges in a hypergraph $H = (V, E)$ requires solving $|\mathcal{E}|$ optimal transport problems where $\mathcal{E}$ is the edge set of the clique expansion $\mathbf{C}(H)$. In practice, the optimal transport costs between two measures with at most $n$ points are computed using the Sinkhorn algorithm from Sinkhorn (1974), which provides an $\varepsilon$-approximation at a cost $O(n^2/\varepsilon^2)$ according to Lin et al. (2019). This implies worst-case complexities of order $O\left(|\mathcal{E}| \times \max_{x \in V}|\mathcal{N}(x)|^2/\varepsilon^2\right)$ for node-Ricci and $O\left(|\mathcal{E}| \times \max_{x \in V}|St(x)|^2/\varepsilon^2\right)$ for edge-Ricci. In turn, large hyperedges have a stronger impact on the node-Ricci complexity.

To justify these theoretical claims, we consider a $K$-uniform hypergraph with 1000 nodes and 300 edges of length $K$ drawn uniformly at random. We report the computational times in seconds for a single Ricci-flow

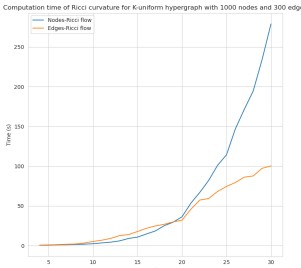

Figure 6: Computation times of Ricci curvature for both methods on a $K$-uniform hypergraph as a function of $K$.

interation in Figure 6. Computations are carried out for unweighted clique and line expansions and are averaged over five different hypergraph initializations. We observe that the computational cost of nodes-Ricci curvature indeed blows up for hypergraphs with large hyperedges due to the inflated neighborhood size $|\mathcal{N}(x)|$. Edge-Ricci flow demonstrates much slower growth in computational time, remaining efficient even for hypergraphs with large hyperedges. Computations are executed on a single 8-core laptop with a Intel(R) Core(TM) i5-8300H CPU @ 2.30GHz processor unit. These computations explain why only edge-Ricci could produce results in a reasonable time on the `Mushroom` dataset. Although this dataset has relatively few hyperedges, the hyperedges are exceptionally large, inducing severe computational challenges for node-Ricci flow. The computational times from Figure 6 along with the quantitative results from Table 2 indicate that Ricci-flow based methods are in particular more suited to dealing with rather small hypergraphs.

### 4.4 Wrap-up, comparison of Node-Ricci and Edge-Ricci flows

Based on the findings from Sections 3.3 and 4, node-Ricci and edge-Ricci flows are complementary approaches for competitive hypergraph clustering. Indeed, as the first one primarily leverages the clique expansion and the second one the line expansion information, taken together, they capture most of the information of the hypergraph. More precisely, edge-Ricci is preferable to node-Ricci whenever:

- There are many small communities.

- The intra-community edges are typically smaller than inter-community edges.

- The hypergraph has very large hyperedges, inducing prohibitive computations on the clique graph.

When dealing with real data with no structural a priori on the communities, both methods tend to perform comparatively, see Section 4.2. However, they can have sensibly different computational costs making one preferable over the other depending on the structure of the hypergraph, in particular the number of hyper-edges and their size. We note that alternating between iterations of node-Ricci flow and edge-Ricci flow did not provide any substantial quantitative benefit.

## Conclusion

We have developed two methods to extend Ricci flow to hypergraphs. The first one, node-Ricci flow, applies standard Ricci flow on the clique expansion and further aggregates it to hyperedges. The other one, the edge-Ricci flow is the main contribution of this article and constitutes an original way to transport edges using the line expansion. Both methods define new weights on the hyperedges, that can then simply be turned into a partitioning algorithm. Each method has its own advantages depending on the specific characteristics of the hypergraph. A first natural extension of this work would be to consider a co-optimal transport of both nodes and edges, see Titouan et al. (2020); Chowdhury et al. (2024). The notion of discrete Ricci flow has more practical uses than graph partitioning, see for instance Khodaei et al. (2024). In particular, it can be used as a method to preprocess hypergraphs by generating new weights that better highlight underlying community structures.

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

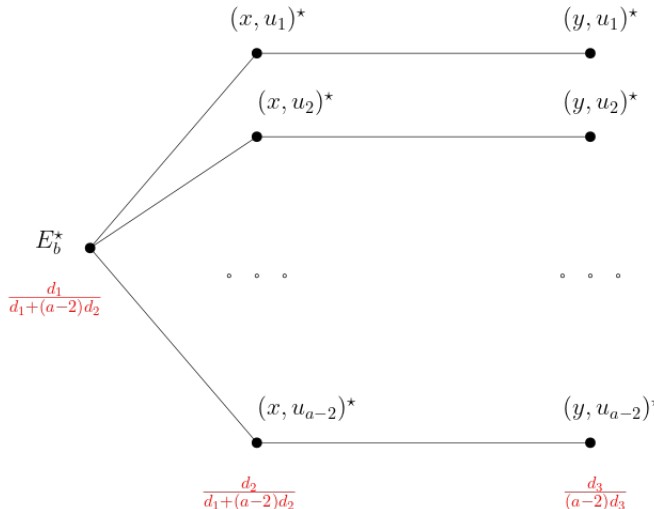

Figure 7: Restriction of the line graph to compute the flow between a gateway and a non-gateway node in Proposition 1.

## A  Proof of Proposition 1

- We start by computing $w_{\mathcal{N},3}^{(l)}$. Let $x$ and $y$ be two non-gateway nodes from the same clique $(x, y, u_1, \ldots, u_{a-2})$. We have that $St(x) \setminus (x, y) = \{(x, u_1), \ldots, (x, u_{a-2})\}$ and $St(y) \setminus (x, y) = \{(y, u_1), \ldots (y, u_{a-2})\}$. The optimal way to transport the uniform probability measure on $St(x) \setminus (x, y)$ to the uniform probability measure on $St(x) \setminus (x, y)$ is to transport each $(x, u_i)$ to $(y, u_i)$, which are adjacent in the line graph. Each edge has weight $1/(a - 2)$ and there is $a - 2$ of them, hence a total cost of 1. Hence by induction, since $w_{\mathcal{N},3}^{(l)} = 1$, we have that $w_{\mathcal{N},3}^{(l)} = 1$ for all $l$.

- For $w_{\mathcal{N},1}^{(l)}$, let $x$ and $y$ be two distinct gateway-nodes, connected by the large hyperedge $E_b$. We have $St(x) \setminus E_b = \{(x, u_1), \ldots, (x, u_{a-1})\}$ and $St(y) \setminus E_b = \{(y, v_1), \ldots, (x, v_{a-1})\}$. The optimal way to transport the uniform probability measure on $St(x) \setminus E_b$ to the uniform probability measure on $St(y) \setminus E_b$ is to transport each $(x, u_i)$ to a corresponding $(y, v_i)$ via $E_b$. This corresponds to a path of length 2 on the line graph. Each edge has weight $1/(a - 1)$ and there is $a - 1$ of them, hence a total cost of 2. As the cost is the same for each pair of nodes in $E_b$, the flow of $E_b$ is equal to that of any pair $(x, y) \in E_b$. We therefore have that $w_{\mathcal{N},3}^{(l)} = 2$ for all $l \geq 1$.

- For $w_{\mathcal{N},2}^{(l)}$, let $x$ be a gateway node and $y$ and non-gateway node from the same clique $(x, y, u_1, \ldots, u_{a-2})$. We have $St(x) \setminus (x, y) = \{E_b, (x, u_1), \ldots, (x, u_{a-2})\}$ and $St(y) \setminus (x, y) = \{(y, u_1), \ldots, (y, u_{a-2})\}$. We represent the line graph restricted to $St(x) \cup St(y) \setminus (x, y)$ in Figure 7. We write the mass of each edge according to Equation 3 where we write $d_i = w_{\mathcal{N},i}^{(l-1)}$ to simplify the notation. We have just demonstrated that $d_1 = 2$ and $d_3 = 1$. The optimal transport plan between $\mu_x$ and $\mu_y$ transports the mass on each $(x, u_i)$ to $(y, u_i)$. These edges are adjacent in the line graph. These $(a - 2)$ edges therefore occupy a fraction of the total cost of $\frac{(a-2)d_2}{2+(a-2)d_2}$. The mass $\frac{d_2}{2+(a-2)d_2}$ on $E_b$ is equally transported to each $(y, u_i)$ through the corresponding $(x, u_i)$. Each $(y, u_i)$ is at a distance 2 from $E_b$, see Figure 7. This implies a cost of $\frac{2d_2}{2+(a-2)d_2}$ for transporting the mass at $E_b$. In the end, the total cost of transporting $\mu_x$ to $\mu_y$ is $\frac{4+(a-2)d_2}{2+(a-2)d_2}$. This implies that

$$w_{\mathcal{N},2}^{(l)} = \frac{4 + (a-2)w_{\mathcal{N},2}^{(l-1)}}{2 + (a-2)w_{\mathcal{N},2}^{(l-1)}}.$$

We can easily prove by induction on $l$ that for every $l$, $w_{\mathcal{N},2}^{(l)} \in [1, 2]$.

