# OpenReview forum: "Hypergraph clustering using Ricci curvature: an edge transport perspective"
_TMLR — Rejected by TMLR_

### Review · Reviewer_6yxN · 2025-03-02

**Summary Of Contributions:**

This paper introduces a novel method for hypergraph clustering by extending Ricci curvature-based techniques traditionally used in graph analysis. The key contribution is the introduction of edge-based Ricci flow, which offers an alternative to the node-based Ricci flow that relies on clique expansions. The proposed approach leverages line expansions, where hyperedges are treated as nodes, enabling more efficient and structure-preserving transport of probability measures. This method is particularly effective for clustering hypergraphs with large hyperedges or small, dense communities, where traditional clique-based methods struggle.

The authors compare node-Ricci flow and edge-Ricci flow, demonstrating that each approach captures different aspects of hypergraph structure. Through extensive experiments on synthetic and real-world datasets, they show that edge-Ricci flow can outperform node-Ricci flow in scenarios where hyperedges span multiple communities or when hypergraph structure is highly non-uniform. Additionally, edge-Ricci flow is computationally more efficient than node-Ricci flow for hypergraphs with large hyperedges, making it more scalable.

**Audience:**

Yes

**Broader Impact Concerns:**

No concerns

**Claims And Evidence:**

Yes

**Requested Changes:**

Some corrections:

1) p.g. 4 Clustεring vol(A), maybe you mean vol($C_i$) ? (MIsmatch in the notation)

2) p.g. 6 J<accard -> Jaccard

3) Some abbliation studies about the role of the $\alpha$ parameter, the number of flow iterations, and the aggregation functions.

4) Explore the sensitivity of the optimal threshold for trimming edges $(\tau)$ across different types of datasets to better understand its impact on clustering performance.

**Strengths And Weaknesses:**

$\textbf{Strengths:}$
1) The introduction of edge-Ricci flow applied to hypergraphs is an original and valuable contribution to the literature. It provides a new perspective on how to utilize Ricci curvature in hypergraph clustering, extending the theory from traditional graph settings to hypergraphs.

2) The paper is well-grounded in Ricci curvature theory, with a detailed explanation of its application to graphs, followed by a natural extension to hypergraphs.

3) The paper is well-structured, with clear explanations of the methods, experiments, and results. The inclusion of a toy example helps readers understand the theoretical implications of the methods before they delve into the experimental sections.

$\textbf{Weaknesses:}$

1) The paper compares the Ricci-based methods to neural network-based methods on real datasets, showing that the proposed method generally underperforms, except for one dataset. In particular, TriCL (not the proposed method) outperforms all other methods (including the proposed one) on three datasets. As a neural network-based approach for hypergraph embeddings, TriCL is more flexible and can be applied to various tasks, such as classification, whereas the proposed method is quite limited in scope, mainly suited for community detection, where its performance is rather underwhelming.

2) In general, the proposed method does not outperform Node-Ricci, even though their performances are similar. However, it is more time-efficient when dealing with large hyperedges.

3) The scalability of the method, particularly with respect to large real-world datasets is a concern, as it is mentioned by the authors. Although the authors acknowledge the computational cost, further exploration into the scalability of edge-Ricci flow in terms of time complexity, memory usage, and performance would be beneficial, especially as hypergraph size and complexity increase.

---

> ### Author Response · Authors · 2025-03-26
> **Response to reviewer 6yxN**
>
> Thank you for the prompt and detailed review. Replying to the weaknesses:
>
> 1°) Indeed, the raw accuracy performance generally fails to be as competitive as TriCL. This is a reason why we chose to publish this work at TMLR, since raw performance does not seem to be a strong acceptance criterion. We also claim that a strength of our method lies in its high level of interpretability as opposed to TriCL which relies on a neural network architecture. Indeed, our paper focuses solely on the clustering problem. However, Ricci flow methods provide a way to generate new weights on the hyperedges sensitive to the community structure. This can be included as a pre-processing step for any further analysis on hypergraphs (including embeddings).
>
> 2°) It is true that for real data, both methods roughly have a similar performance. However, on synthetic data, edge-Ricci highly outperforms node-Ricci if there are large cross-community edges. Concerning the time efficiency, it becomes a critical concern for node-Ricci, making the computation unfeasible if there are too many very large hyperedges. I believe this is a general remark going beyond the present clustering problem regarding hypergraphs. Indeed, turning them into their clique graphs as commonly done in applications, as the can cause the complexity of the object to blow up. I will try to make a stronger point out of it.
>
> 3°) I can include more details about the memory usage. However, I am not sure to grasp the difference between "computational cost", "time complexity" and "performance". Do you have something specific in mind?
>
> Replying to the requested changes:
> 1-2°) Thanks for pointing this out
> 3°) Concerning the alpha parameter and the number of flow iterations, these parameters are inherited from standard Ricci curvature on graphs and have already been discussed. I will point out more specific references, but I do not think that a thorough analysis of them is necessary when generalizing to hypergraphs. I will add the numerical results on real data with other aggregation functions in the appendix.
> 4°) This is a good point, I am thinking of adding a comparative experiment of the performance for:
> *The optimal tau that gives the best clustering accuracy
> *The tau that minimizes the hypergraph modularity
> *The tau that minimizes the graph modularity computed on the clique graph
> I believe this can be an interesting addition to the paper.

---

### Review · Reviewer_rzN9 · 2025-03-10

**Summary Of Contributions:**

The paper introduces we introduce a novel method for extending Ricci flow to hypergraphs, in order to define clustering/community detection in hypegraphs. The authors propose a novel definition of Ricci curvature for hypergraphs by considering probability distributions defined on edges, rather than on nodes which is rather interesting and novel. This formulation leverages the hypergraph's line expansion, a graph-based representation in which nodes correspond to hyperedges, and edges represent intersections among these hyperedges. The paper adopts an extensive experimental set-up.

**Audience:**

Yes

**Broader Impact Concerns:**

I do not find any major ethical implications that would require adding a Broader Impact Statement.

**Claims And Evidence:**

Yes

**Requested Changes:**

I would appreciate if the authors would:

- Clearly document hyperparameter tuning procedures.

- Adopt theoretical frameworks (if possible) or introduce some empirical stability analyses.

- Consider robustness analysis by varying parameters systematically.

- Evaluate aggregation robustness across multiple datasets.

This would help remove any concerns against the heuristic nature of the method.

**Strengths And Weaknesses:**

The paper strengths are:

- The paper Introduces a new notion of Ricci curvature explicitly designed for hypergraphs by defining probability distributions on edges rather than nodes.

- In addition, the paper utilises prior theoretical results from Kirkland, 2018 establishing that line expansions retain significant structural information of hypergraphs, thus justifying this representation.

- The paper defines various experiments comparing the proposed edge-transport-based approach against the standard node-transport-based approach and finds comparable/competitive results.

- The proposed method demonstrates the significant computational advantages of edge-based measure transport, particularly in handling hypergraphs with large hyperedges, making the proposed method highly scalable and practical for real-world applications.


The main drawbacks of the method are:

- While Ricci flow itself is a mathematically rigorous geometric approach, its discrete formulation used in this method is heuristic in nature. The proposed approach thus involves several hyperparameters, such as the number of flow iterations and the trimming threshold. Additionally, it relies on modularity for final cluster validation—a measure that is itself heuristic—further compounding this limitation.

- Aggregating pairwise curvatures ignores genuine multi-way relationships intrinsic to hyperedges. Hypergraphs explicitly capture relationships involving more than two nodes, and simply aggregating pairwise measures may not fully reflect this higher-order complexity. The choice of aggregation function may heavily affects the outcome.

- The proposed Ricci curvature heavily depends on how you define the probability measures , e.g., uniform vs. intersection-based measures. Where different choices may drastically change curvature values and clustering outcomes, potentially making results sensitive and unstable.

- In addition, while transporting measures defined on edges instead of nodes is novel, but it also moves interpretation away from node-level interactions to edge-level structures, making direct interpretation less intuitive. With the authors arguing for the opposite.

- The proposed curvature definition is novel and, as such, lacks theoretical guarantees (e.g., convergence, consistency). The method dependent on parameter α but the choice and effect of this parameter is not clearly established. I.e. why the authors propose only two choices for α? are there any other alternatives? The authors also state "We did not observe a great sensibility to the α parameter for the measures µ defined on the edges" but these results are not provided in the manuscript.

- The results are not so convincing to argue that it outperrforms the competiting methods. I would also argue that many of these baselines are not state-of-the-art, e.g. Node2vec and DGI can be considered as more of classical approaches.

---

> ### Author Response · Authors · 2025-03-26
> **Response to reviewer rzN9**
>
> Thank you for your prompt and thorough review. Replying to your listed drawbacks and requested changes:
>
> Indeed it is mostly heuristic, especially for the edge-transport version. There is indeed a lot of hyperparameters, some being more crucial than others, and your concerns seem to be mostly related to that. Let me go through the hyperparameters and add some comments:
>
> -Concerning the alpha parameter and the number of flow iterations, these parameters are inherited from standard Ricci curvature on graphs and have already been discussed. I will point out more specific references, but I do not think that a thorough analysis of them is necessary when generalizing to hypergraphs. The choices of alpha we gave were for the sake of explanations/theoretical guarantees, but of course, any parameter alpha between 0 and 1 can be considered.
>
> -Concerning the trimming threshold and the modularity function, this approach yields a hierarchical clustering, where tau determines where to cut in the hierarchy. Using the modularity only aims at comparing with a given baseline but in practice, several choices of tau could be considered and be valid.
> I am thinking of adding an additional ablation experiment comparing the performance between:
>
> *The optimal tau that gives the best clustering accuracy
>
> *The tau that minimizes the hypergraph modularity
>
>  *The tau that minimizes the graph modularity computed on the clique graph I believe this can be an interesting addition to the paper.
>
> -Concerning the aggregation function. We adopted an approach similar to the paper "Ollivier-Ricci Curvature for Hypergraphs: A Unified Framework" (https://arxiv.org/abs/2210.12048). Indeed, aggregating pairwise information probably incurs some loss of information. Any thoughts on how to improve this are more than welcome. We did notice some sensitive differences between different aggregation metrics, taking the max always being preferable. I will add the numerical results in Appendix.
>
> -Concerning the way to define probability measures (uniform/intersection/Jaccard), although it could theoretically induce a lot of differences, we noticed a very similar performance on real data. I will add the numerical results in Appendix.
>
> I will include more details about these hyperparameters in the paper as well.
>
> Regarding "In addition, while transporting measures defined on edges instead of nodes is novel, but it also moves interpretation away from node-level interactions to edge-level structures, making direct interpretation less intuitive." that is perfectly true and we do point this out at the bottom of Page 7. We do not argue that edge-Ricci is more interpretable and intuitive, simply more "respectful" of the hypergraph structure in presence of large hyperedges. I will make sure that this is clear throughout the paper to remove any ambiguity.
>
> I wish some theoretical guarantees could be derived, but they seem currently out of reach. The theory for hypergraph is my opinion quite underdeveloped as opposed to graphs.
>
> The numerical results are indeed quite humble compared to the most competitive methods (which does not seem to be a strong decision point for TMLR). As such, I think it is important to include more classical approaches to see how our method compares with them. I will say a word about it.
>
> I hope this clarifies things and the intended changes respond to your concerns.

---

### Review · Reviewer_uoad · 2025-03-25

**Summary Of Contributions:**

This paper proposes a method for extending Ricci flow to hypergraphs by defining probability measures on the edges and transporting them on the line expansion. This approach yields a new weighting on the edges, which proves particularly effective for community detection. Authors compare their approach against Ricci flow defined on the clique expansion in terms of sensitivity to the hypergraph structure.

**Audience:**

Yes

**Broader Impact Concerns:**

N/A - this work is investigating on public benchmark datasets for hypergraph clustering without the involvement of human subjects.

**Claims And Evidence:**

Yes

**Requested Changes:**

See above section for details: (1) revision introduction, (2) discussion on architectural design choice and motivation, (3) time complexity analysis.

**Strengths And Weaknesses:**

The Introduction section is missing some references to important relevant work about a model design to predict social interactions. The authors should include this in the discussion for recency.

[NeurIPS 2024] “Non-Euclidean Mixture Model for Social Network Embedding” Part of Advances in Neural Information Processing Systems 37 (NeurIPS 2024) Main Conference Track. https://proceedings.neurips.cc/paper_files/paper/2024/hash/c9e20f70f049ac5be8955c8bb970d0a5-Abstract-Conference.html

As an alternative to Ricci curvature, how do other curvature methods like hyperbolic space (negative curvature) or spherical space (positive curvature) compare in terms of effectiveness in hypergraph clustering? This topic would be useful in the paper discussion.

Can the authors include a discussion on the time complexity of their approach?

---

> ### Author Response · Authors · 2025-03-26
> **Response to Reviewer uoad**
>
> Thank you for the review.
>
> Thank you for pointing out this additional reference, I will mention it in the paper's Introduction. Indeed, several notions of curvature exist beyond Ollivier-Ricci curvature, some of which have been adapted to graphs. Further extending them to hypergraphs constitutes a separate research topic, I will add a word about it in the discussion.
>
> I did include a time complexity analysis (Section 4.3). Did you have something else in mind?

---

> > ### Comment · Reviewer_uoad · 2025-04-24
> >
> > Thank you. Looking forward to seeing the new version with the revisions.

---

### Decision · Action_Editor_Da3d · 2025-05-03

**Recommendation:** Reject

**Comment:**

The paper was reviewed by three reviewers. The reviewers raised concerns about the effectiveness of the proposed method, its scalability, its dependence on a large number of hyperparameters, the limitations of the employed functions that aggregate curvatures of pairs of nodes, missing related work, and missing baselines. More importantly, two reviewers requested further ablation studies that would investigate the effects of the different hyperparameters and the robustness of the proposed method. The authors responded to the reviewers' comments, and promised to extend the experimental section, thus addressing the above concerns. However, no revised manuscript was submitted by the authors along with the response. The overall recommendations from the reviewers were mixed, with two reviewers advocating for weak acceptance and another advocating for rejection.

Overall, I like the direction of the paper. However, I also agree with the reviewers that the evaluation is not thorough enough, and that some ablation studies would significantly strengthen the paper and would provide insights for the better understanding of the key aspects of the work. I also find some of the claims made in the paper problematic. For instance, claims about the explainability of the proposed method are not supported by convincing evidence. These concerns prevent me from recommending acceptance of this paper in its current form. The paper would greatly benefit from a revision of the actual claims and empirical results.

**Audience:**

Community detection is a fundamental problem in machine learning with practical significance. Therefore, the findings of this paper will be of interest to some individuals in TMLR's audience.

**Claims And Evidence:**

This paper deals with the problem of detecting communities in hypergraphs. A new method is proposed which generalizes Ricci flow to hypergraphs. The method uses the line expansion of the hypergraph to transport measures defined on the edges. It is empirically compared mainly against a similar approach which applies Ricci flow on the clique expansion and aggregates it to hyperedges. The authors claim that the proposed method is more sensitive to the hypergraph structure, especially in hypergraphs that contain large hyperedges. This claim is supported by empirical evidence, but the experiments only focus on a specific type of hypergraphs. Another claim made in the submission is that the proposed method exhibits a higher level of explainability compared to approaches based on neural network embeddings. However, this claim is not supported by clear evidence.

**Resubmission Of Major Revision:**

The authors may consider submitting a major revision at a later time.